# Corin Deficiency Diminishes Intestinal Sodium Excretion in Mice

**DOI:** 10.3390/biology12070945

**Published:** 2023-07-01

**Authors:** Xiabing Gu, Kun Wang, Wenguo Li, Meiling He, Tiantian Zhou, Meng Liu, Qingyu Wu, Ningzheng Dong

**Affiliations:** 1Cyrus Tang Hematology Center, Collaborative Innovation Center of Hematology, State Key Laboratory of Radiation Medicine and Prevention, Soochow University, Suzhou 215123, China; xiabinggoouu@163.com (X.G.); wkun1207@163.com (K.W.); liwenguoguigui@163.com (W.L.); meilinghe8@163.com (M.H.); ttzhou@suda.edu.cn (T.Z.); liumeng@suda.edu.cn (M.L.); 2NHC Key Laboratory of Thrombosis and Hemostasis, Jiangsu Institute of Hematology, Medical School, Suzhou 215006, China

**Keywords:** ANP, corin, intestine, mouse model, sodium excretion, sodium homeostasis, transmembrane serine protease

## Abstract

**Simple Summary:**

Sodium homeostasis is critical for body fluid balance. The hormonal regulation of sodium reabsorption and excretion is central in sodium homeostasis. The kidney is the main organ that controls sodium reabsorption and excretion. Corin is a proteolytic enzyme responsible for activating atrial natriuretic peptide (ANP), a key hormone that promotes renal natriuresis. The heart is the main organ for corin and ANP expression. In this work, we report that corin and ANP are expressed in the mouse intestine, where sodium excretion also occurs. Corin deficiency impairs fecal sodium excretion in mice. These findings reveal a new function of corin in the intestinal tract to promote sodium excretion.

**Abstract:**

Sodium excretion, a critical process in sodium homeostasis, occurs in many tissues, including the kidney and intestine. Unlike in the kidney, the hormonal regulation of intestinal sodium excretion remains unclear. Atrial natriuretic peptide (ANP) is a crucial hormone in renal natriuresis. Corin is a protease critical for ANP activation. Corin and ANP are expressed mainly in the heart. In this study, we investigated corin, ANP, and natriuretic peptide receptor A (Npra) expression in mouse intestines. Corin and ANP expression was co-localized in enteroendocrine cells, whereas Npra expression was on the luminal epithelial cells. In *Corin* knockout (KO) mice, fecal Na^+^ and Cl^−^ excretion decreased compared with that in wild-type (WT) mice. Such a decrease was not found in conditional *Corin* KO mice lacking cardiac corin selectively. In kidney conditional *Corin* KO mice lacking renal corin, fecal Na^+^ and Cl^−^ excretion increased, compared to that in WT mice. When WT, *Corin* KO, and the kidney conditional KO mice were treated with aldosterone, the differences in fecal Na^+^ and Cl^−^ levels disappeared. These results suggest that intestinal corin may promote fecal sodium excretion in a paracrine mechanism independent of the cardiac corin function. The increased fecal sodium excretion in the kidney conditional *Corin* KO mice likely reflected an intestinal compensatory response to renal corin deficiency. Our results also suggest that intestinal corin activity may antagonize aldosterone action in the promotion of fecal sodium excretion. These findings help us understand the hormonal mechanism controlling sodium excretion the intestinal tract.

## 1. Introduction

Sodium is an electrolyte essential for normal cellular function and body fluid balance [1]. Impaired sodium homeostasis contributes to major diseases, such as hypertension, stroke, and chronic kidney disease [2,3,4,5]. A fine balance of sodium and body fluid is controlled by complex behavioral and hormonal mechanisms [1,4,6]. In the kidney, the primary organ that controls sodium homeostasis, the adrenal-gland-derived aldosterone enhances sodium and water reabsorption, whereas atrial natriuretic peptide (ANP), a hormone made mostly in the heart, promotes sodium and water excretion [7,8,9,10]. The equilibrium of sodium (re)absorption and excretion is central in sodium homeostasis.

Corin is an essential proteolytic enzyme in ANP activation [11,12]. It contains an N-terminal single-span transmembrane domain, which is common in type II transmembrane serine proteases [13,14,15,16]. Like ANP, corin expression is mostly in the heart [13,17,18]. In mouse models, corin deficiency prevents ANP generation, reduces renal sodium excretion, and causes hypertension and cardiac dysfunction [19,20,21,22,23]. In longitudinal human genetic studies, *CORIN* variants are linked to dietary salt sensitivity, blood pressure variations, and risk of hypertension [24]. Consistently, dysfunctional variants in the *CORIN* gene are associated with hypertension, coronary artery disease, and heart failure [25,26,27,28,29]. Altered levels of circulating corin from ectodomain shedding are also found in individuals with cardiovascular and metabolic diseases [30,31,32,33,34,35].

Low levels of corin expression have been reported in non-cardiac tissues, e.g., the uterus [36,37,38], kidney [18,39,40], skin [41,42], and bone [43,44]. In the uterus, reduced corin and ANP expression impairs endometrium decidualization and spiral artery remodeling, which are important for healthy pregnancy [45,46]. In the kidney and dermal sweat glands, corin-mediated ANP production stimulates sodium excretion [42,47]. To date, decreased corin levels have been reported in animal models of kidney disease and in humans with chronic renal failure [39,48,49,50]. These findings indicate a critical role of corin in the control of sodium homeostasis and vascular biology in diverse tissues.

The intestine is an important organ where both sodium uptake and excretion occur [51,52,53]. Most sodium uptake is in the small intestine, whereas sodium excretion takes place mainly in the colon. Compared to what is known in the kidney, the hormonal regulation of sodium homeostasis in the intestine is less well defined. Since corin enhances sodium excretion in the kidney and skin eccrine sweat glands [23,42], it is conceivable that corin may have a similar function in the intestine to promote sodium excretion. In this study, we tested this hypothesis by analyzing corin, ANP, and natriuretic peptide receptor A (Npra) expression in mouse intestinal segments. Additionally, we examined intestinal sodium and chloride excretion in wild-type (WT) and *Corin* knockout (KO) mouse models with dietary salt challenge. Our results should help elucidate a potential function of corin in the intestine.

## 2. Materials and Methods

### 2.1. Mouse Models

Strains of *Corin*-deficient mice, including global KO (*Corin* KO) [42], heart conditional KO (hcKO) [42,47], and kidney conditional KO (kcKO) [47] mice, were published previously. Briefly, two loxP sites were inserted into the *Corin* gene flanking exon 4 to generate *Cor^flox^* mice [42]. *Corin* KO and hcKO mice were created by crossing *Cor^flox^* mice with *CMV*-*Cre* mice (B6.C-Tg(CMV-cre)1 Cgn/J) with ubiquitous *Cre*, and *Myh6-Cre* mice (B6.FVB(129)-Tg(Myh6-cre/Esr*)1 Jmk/J) with cardiac *Cre*, respectively (42). *Corin* kcKO mice were created by crossing *Cor^flox^* mice with *Ggt1-Cre* mice (Tg(Ggt1-cre)M3Egn/J) with renal *Cre* [47]. The mice were crossed (more than 10 generations) into the C57BL/6J background. As reported previously [47], *Corin* KO, hcKO, and kcKO mice were hypertensive. When on a 0.3% NaCl diet, levels of systolic blood pressure were comparable among these three strains of mice, but all higher than that in WT mice [47]. When on a 4% NaCl diet, *Corin* KO and kcKO mice, but not hcKO mice, exhibited salt-exacerbated hypertension [47].

WT C57BL/6 and the *Corin*-deficient mice were kept at a certified animal facility with 12 h light cycles and free access to water and chow diet (casein 200 g/kg, L-cystine 3 g/kg, cornstarch 397.5 g/kg, maltodextrin 132 g/kg, sucrose 100 g/kg, cellulose 50 g/kg, soybean oil 70 g/kg, mineral mix 35 g/kg, vitamin mix 10 g/kg, and choline bitartrate 2.5 g/kg) with normal (0.3% NaCl) or high (4% NaCl) salt contents (Suzhou Shuangshi Laboratory Animal Feed Science, Suzhou, China). Mice (6–12 weeks old, depending on the experiments) were randomized into test groups. The procedures were conducted with an approved protocol (202008A701; the Animal Use and Care Committee of Soochow University).

### 2.2. Tissue Collection and Preparation

Tissues, including the heart, kidney, liver, and intestine, were isolated from mice (male, 10–12 weeks old). Intestinal segments (duodenum, jejunum, ileum, and colon) were separated. Each tissue sample was divided into two parts; one was fixed in 4% paraformaldehyde at room temperature for hematoxylin and eosin (H&E) and immune staining experiments and the other was stored in liquid nitrogen and used later for gene expression studies. Samples from 5–8 mice and at least 20 sections were included in each test group.

### 2.3. Immunostaining

Mouse intestinal tissues were treated with paraformaldehyde (4%) and embedded in paraffin. Serial 4 μm sections were prepared using microtome (RM2245, Leica, Nussloch, Germany) and arranged on slides (188105, Citotest Scientific, Haimen, China). The paraffin-embedded sections were treated with xylene and graded ethanol solution, as described previously [54]. For immunohistochemistry (IHC), the sections were incubated with primary antibodies against corin (1:500, ab255812, Abcam, Cambridge, UK), pro-ANP/ANP (1:300, GTX109255, GeneTex, Irvine, CA, USA), Npra (1:500, GTX109810, GeneTex), chromogranin A (ChgA) (1:500, ab254322, Abcam), and mucin 2 (Muc2) (1:2000, ab272692, Abcam). As a negative control, the primary antibody was replaced by immunoglobulin G (IgG) from normal mouse serum (I8765, 1:1000, Sigma, St Louis, MO, USA). After incubation at 4 °C for 12 h, all sections were washed and incubated with a horseradish peroxidase-conjugated detecting antibody (MaxVision, kit-5005, Maxim Biotechnologies, London, UK). After 30 min at 37 °C, the sections were washed, counterstained with hematoxylin, and inspected under a camera-equipped microscope (Leica DM2000 LED; Wetzlar, Germany).

To co-localize proteins in intestinal tissues, sections were prepared, as described above, and incubated with primary antibodies against corin (1:50, ab255812, Abcam), pro-ANP/ANP (1:50, ab190001, Abcam), NPR-A (1:200, GTX109810, GeneTex), ChgA (1:100, ab254322, Abcam) (an enteroendocrine cell marker) [55], Muc2 (1:500, ab272692, Abcam) (a goblet cell marker) [56], and E-cadherin (E-cad) (1 μg/mL, ab231303, Abcam). After washing, the sections were incubated with an Alexa-488 (green)-, 594 (red)-, or 647 (purple)-labeled detecting antibody (1:500, A21202 and A11012, Invitrogen, Carlsbad, CA, USA) (1:500, ab150131, Abcam). After additional washing, the slides were stained with 4′,6-diamidino-2-phenylindole (DAPI) (0100–20, Southern Biotech, Birmingham, AL, USA) for cell nuclei and inspected under a confocal microscope (Olympus, FV1000, Tokyo, Japan).

### 2.4. Gene Expression Analysis by PCR and qRT-PCR

RNA samples were isolated from tissues with a Trizol kit (15596018, Thermo Fisher Scientific, Waltham, MA, USA) and used to make cDNAs (K1622, Thermo Fisher Scientific). *Corin*, *Nppa* (encoding prepro-ANP), and *Npra* mRNA-derived fragments were amplified by PCR and separated by agarose gel electrophoresis. To assess gene expression levels, quantitative (q) RT-PCR was conducted using the QuantStudio 6 Real-Time PCR System (Thermo Fisher Scientific). The data were normalized with *Gapdh* (glyceraldehyde 3-phosphate dehydrogenase) levels. Appendix A lists oligonucleotide primer sequences used in the PCR and qRT-PCR experiments.

### 2.5. Aldosterone Treatment

Effects of aldosterone on fecal Na^+^ and Cl^−^ excretion were tested in mice on 0.3% NaCl or 4% NaCl diet. After 3 weeks on the normal or high-salt diet, the mice were anesthetized with ketamine (40 mg/kg of body weight) (K165300, Toronto Research Chemicals, Toronto, Canada) and xylazine (2.5 mg/kg of body weight) (B3344, APExBIO, Houston, TX, USA). An osmotic pump (1001 W, RWD Life Science, Shenzhen, China) with aldosterone diluted in 5% alcohol (3.5 μg/day, for 3 days) (C4453, APExBIO) was inserted subcutaneously. After the surgery, the mice were returned to their cages. Fecal samples were collected. Fecal Na^+^ and Cl^−^ levels were analyzed, as described below.

### 2.6. Fecal Na^+^ and Cl^−^ Measurements

WT, *Corin* KO, hcKO, and kcKO mice (male, n = 5–10 per group) with or without the aldosterone treatment were placed singly in metabolic cages (Fengshi Laboratory Animal Equipment, Suzhou, China) with 0.3% or 4% NaCl diet and free access to water. Fecal samples on three consecutive days were gathered. The samples were dried at 80 °C for 24 h, weighed, and redissolved in distilled water (0.2 g/2 mL) at 4 °C. After 24 h, the samples were centrifuged at 12,000 g for 20 min. The supernatant was collected and Na^+^ and Cl^−^ concentrations were determined using an electrolyte instrument (6230 M, Jenco Instruments, San Diego, CA, USA).

### 2.7. Statistics

Data were examined using Prism 8.0 software (GraphPad, San Diego, CA, USA). All data were analyzed for the normality using Anderson–Darling, Shapiro–Wilk, Kolmogorov–Smirnov, and D’Agostino–Pearson tests. Unpaired Student *t* test (two-tailed) was used to compare the data between two groups. One-way analysis of variance (ANOVA) and Turkey’s post-test or two-way ANOVA and Turkey or Bonferroni multiple comparison tests were used for multiple groups. Data are presented as mean ± SEM. A *p* value < 0.05 was considered significant.

## 3. Results

### 3.1. Corin Protein Expression in Mouse Intestines

To examine corin protein expression in intestinal segments, we performed IHC experiments using serial intestinal sections from WT mice. We included antibodies against ChgA (an enteroendocrine cell marker) and Muc2 (a goblet cell marker). Corin staining was detected in jejunal sections (Figure 1, second column in rows 1 and 2). The expression pattern and cellular location resembled those of ChgA staining (Figure 1, third column in rows 1 and 2), but not those of Muc2 staining (Figure 1, fourth column in rows 1 and 2). In negative controls, in which the primary antibody was replaced by normal IgG, no positive staining was observed (Figure 1, right column in rows 1 and 2). Similar corin staining was observed in colonic sections, which also overlapped with ChgA, but not Muc2, staining (Figure 1, 2nd–4th columns in rows 3 and 4). These results indicate that corin is likely expressed in enteroendocrine cells in the mouse intestinal tract.

### 3.2. Intestinal Corin, Nppa, and Npra mRNA Expression in WT and Corin KO Mice

To verify our findings, we examined *Corin*, *Nppa* (encoding prepro-ANP), and *Npra* (encoding natriuretic peptide receptor A) mRNA expression in intestinal segments of WT mice. In RT-PCR, *Corin* mRNA was detected in samples from nearly the entire intestinal tract, including the duodenum, jejunum, ileum, cecum, and colon, from WT mice (Figure 2A, first row in left panel). *Corin* mRNA expression was found in heart and kidney (positive controls) but not liver (negative control) samples from WT mice. As another negative control, no *Corin* mRNA expression was found in heart, kidney, and intestine samples from *Corin* KO mice (Figure 2A, first row in right panel). In parallel experiments, *Nppa* mRNA was found in all the tissues tested, except the liver, from WT and *Corin* KO mice (Figure 2A, second row in both panels), whereas *Npra* mRNA expression was observed in all the tissues tested from WT and *Corin* KO mice (Figure 2A, third row in both panels). We quantified *Corin*, *Nppa,* and *Npra* mRNA levels in mouse colon samples via qRT-PCR. The result showed minimal colonic *Corin* mRNA levels in *Corin* KO mice, compared with that in WT mice (Figure 2B, left panel). In contrast, similar *Nppa* and *Npra* mRNA levels were found in colons from WT and *Corin* KO mice (Figure 2B, middle and right panels).

### 3.3. Intestinal Pro-ANP/ANP and Npra Protein Expression in WT and Corin KO Mice

We next analyzed pro-ANP/ANP and Npra protein in jejunal and colonic sections from WT and *Corin* KO mice using IHC (the antibody used in immunostaining could not distinguish ANP from pro-ANP. To simplify the description, we used the term ‘ANP’, instead of ‘pro-ANP/ANP’, in the following sections). Consistent with the findings in RT-PCR, we observed ANP staining in jejunal (Appendix A) and colonic (Figure 3A) samples from WT and *Corin* KO mice. Like in corin staining, ANP and ChgA staining overlapped in enteroendocrine, but not Muc2-positive goblet, cells in the intestinal sections from WT and *Corin* KO mice.

Similarly, Npra staining was found in the jejunal (Appendix A) and colonic (Figure 3B) samples from WT and *Corin* KO mice. Unlike in corin and ANP staining, which was exclusively in the ChgA-positive enteroendocrine cells, Npra staining was mostly on the luminal epithelial surface with minimal overlapping with ChgA and Muc2 staining in enteroendocrine and goblet cells, respectively.

### 3.4. Co-Localization of Corin and ANP Proteins in Intestinal Enteroendocrine Cells

To verify the apparent co-localization of corin and ANP proteins in enteroendocrine cells, we conducted co-immunofluorescent staining of corin, ANP, and ChgA in intestinal samples from WT and *Corin* KO mice. We observed co-staining of corin, ANP, and ChgA in jejunal (Appendix A, first row) and colonic (Figure 4A, first row) sections from WT mice. Similar co-staining of ANP and ChgA, but not corin, was observed in jejunal (Appendix A, second row) and colonic (Figure 4A, second row) sections from *Corin* KO mice. In parallel experiments, Npra staining was observed mostly on the luminal epithelial surface in the jejunal (Appendix A) and colonic (Figure 4B) sections from WT and *Corin* KO mice. These results are in line with findings from the IHC experiments, suggesting an intestinal function of the corin and ANP pathway in the intestinal tract.

### 3.5. Reduced Fecal Na^+^ and Cl^−^ Excretion in Corin KO Mice

The primary function of corin is to activate ANP, thereby enhancing renal sodium excretion. The colon, particularly its distal segment, is another site where sodium excretion occurs. To determine if corin has a role in intestinal sodium excretion, we placed WT and *Corin* KO mice on 0.3% and 4% NaCl diets. After three weeks, jejunum and colon tissues were collected and examined histologically. In H&E-stained sections, no discernible pathological alternations, such as epithelial lesion, cellular disarray, or inflammation, were observed in WT and *Corin* KO mice (Figure 5A,B). Additionally, we found similar dry fecal weight to body weight ratios between WT and *Corin* KO mice on 0.3% NaCl diet (Figure 5C). The fecal weights remained similar between the two groups after the mice were put on a 4% NaCl diet (Figure 5C).

We next measured fecal Na^+^ and Cl^−^ concentrations in WT and *Corin* KO mice. Reduced fecal Na^+^ (Figure 5D) and Cl^−^ (Figure 5E) levels were found in *Corin* KO mice with a 0.3% NaCl diet, compared to those in WT mice with the same diet. When WT and *Corin* KO mice were on a 4% NaCl diet, fecal Na^+^ and Cl^−^ levels increased in the both groups (Figure 5D,E). Compared to those in WT mice, fecal Na^+^ and Cl^−^ levels remained lower in *Corin* KO mice (Figure 5D,E). These data indicate that corin deficiency reduced intestinal Na^+^ and Cl^−^ excretion in mice.

Epithelial sodium channel (ENaC) and cystic fibrosis transmembrane conductance regulator (CFTR) are key mediators in Na^+^ and Cl^−^ reabsorption. We analyzed colon *Scnn1a* (encoding αENaC subunit) and *Cftr* mRNA levels in WT and *Corin* KO mice. By qRT-PCR, we found comparable levels of *Scnn1a* and *Cftr* mRNA levels between WT and *Corin* KO mice (Appendix A). These data show that corin deficiency did not significantly alter colonic *Scnn1a* and *Cftr* mRNA expression in mice.

### 3.6. Cardiac Corin Is Dispensable for Intestinal Na^+^ and Cl^−^ Excretion in Mice

Corin is expressed primarily in the heart, where it serves as a key component in the ANP-mediated endocrine function. Emerging evidence demonstrates that corin also acts outside of the heart, e.g., in the kidney and sweat glands, to promote sodium excretion via an auto/paracrine mechanism. To examine if cardiac corin enhances intestinal Na^+^ and Cl^−^ excretion, we analyzed *Corin* hcKO mice without cardiac corin expression. We found that fecal Na^+^ and Cl^−^ levels in *Corin* hcKO mice were similar to those in WT mice, but higher than those in *Corin* KO mice when the mice were on a 0.3% NaCl diet (Figure 6A,B). When the mice were on a 4% NaCl diet, fecal Na^+^ and Cl^−^ excretion in *Corin* hcKO mice increased. The levels were higher than those in *Corin* KO mice but similar to those in WT mice (Figure 6A,B). These data indicate that cardiac corin is dispensable for fecal Na^+^ and Cl^−^ excretion in mice, suggesting a local corin function in the intestinal tract to promote Na^+^ and Cl^−^ excretion.

### 3.7. Renal-Corin-Deficient Mice Exhibited Increased Fecal Na^+^ and Cl^−^ Excretion

We next examined the role of renal corin in intestinal Na^+^ and Cl^−^ excretion using *Corin* kcKO mice lacking renal corin expression. These mice had impaired urinary Na^+^ and Cl^−^ excretion. In *Corin* kcKO mice on a 0.3% NaCl diet, fecal Na^+^ and Cl^−^ levels were higher than those of WT and *Corin* KO mice on the same diet (Figure 6C,D). When the mice were on a 4% NaCl diet, fecal Na^+^ and Cl^−^ levels increased in all three groups. Consistently, fecal Na^+^ and Cl^−^ levels in *Corin* kcKO mice were high, compared with those in WT and *Corin* KO mice (Figure 6C,D). These data show that renal corin is unnecessary for fecal Na^+^ and Cl^−^ excretion and that intestinal corin may serve as a compensatory mechanism in response to reduced urinary Na^+^ and Cl^−^ excretion caused by renal corin deficiency.

### 3.8. Effects of Aldosterone on Fecal Na^+^ and Cl^−^ Excretion

Aldosterone is a primary hormone that promotes sodium reabsorption in the kidney and colon. To determine if corin promotes salt excretion in the intestine by antagonizing aldosterone function, we treated WT, *Corin* KO, and *Corin* kcKO mice with aldosterone and analyzed fecal Na^+^ and Cl^−^ levels. In the group on a 0.3% NaCl diet, fecal Na^+^ (Figure 7A) and Cl^−^ (Figure 7B) levels decreased in WT and *Corin* kcKO, but not *Corin* KO, mice after the aldosterone treatment, resulting in similar fecal Na^+^ and Cl^−^ levels among the three groups. As described above, fecal Na^+^ and Cl^−^ levels increased when the mice were on a 4% NaCl diet. After the aldosterone treatment, fecal Na^+^ and Cl^−^ levels also became similar among the three groups (Figure 7A,B). These findings suggest that intestinal corin may play a critical role in promoting fecal Na^+^ and Cl^−^ excretion by antagonizing aldosterone action.

## 4. Discussion

Sodium homeostasis is vital for health. Corin-mediated ANP activation is crucial in the regulation of sodium homeostasis [12,57,58]. Given the role of the intestine in sodium uptake and excretion, we examined corin expression along the intestinal tract in mice. Using RT-PCR, we detected *Corin* mRNA in all intestinal segments examined, from the duodenum to the distal colon. Unlike in the heart and kidney, *Corin* mRNA levels in the intestinal segments were low. The immunostaining of corin protein in jejunal and colonic sections revealed that corin was expressed selectively in ChgA-positive enteroendocrine cells, but not in Muc2-positive goblet cells or luminal epithelial cells. This could explain the low mRNA levels in the intestinal samples, compared to those in heart and kidney samples, analyzed in our experiments. As a type II transmembrane protein, corin is tethered on the cell surface. The intestinal expression suggested that corin may play a role in the intestinal tract.

In support of this hypothesis, we found *Nppa* and *Npra* mRNA and pro-ANP/ANP and Npra protein expression in mouse intestinal segments in similar experiments. Importantly, corin and pro-ANP/ANP expression was co-localized in the ChgA-positive enteroendocrine cells. ChgA is a secretory protein found in virtually all enteroendocrine cells along the gastrointestinal tract and in neuroendocrine cells of other tissues [55]. It remains to be determined if corin and pro-ANP/ANP expression is restricted to a specific subset of the intestinal enteroendocrine cells. In the heart, corin and pro-ANP are co-expressed in cardiomyocytes [59,60]. When pro-ANP is released from cardiomyocytes, corin conv erts pro-ANP to ANP on the cell surface [61]. It is probable that a similar cellular mechanism exists in the intestinal enteroendocrine cells, where corin activates ANP, which in turn stimulates Npra on neighboring epithelial cells, triggering downstream signaling to promote sodium excretion. Consistently, we found low levers of fecal sodium and chloride in *Corin* KO mice, compared to those in WT mice, on normal- and high-salt diets. These data suggest that corin, via ANP activation, is important for promoting sodium and chloride excretion in the intestine.

In the heart, corin and ANP are key components of the cardiac endocrine system. Latest studies indicate that corin and ANP also act in the kidney and eccrine sweat glands to promote sodium excretion via an auto/paracrine mechanism [42,47]. In *Corin* kcKO mice lacking renal, but not cardiac, corin, impaired urinary sodium excretion and worsening hypertension were observed upon dietary salt overload, an indication that circulating ANP originating from cardiac corin probably failed to compensate renal corin deficiency when dietary salt intakes were high [47]. In this study, decreased fecal Na^+^ and Cl^−^ levels were found in *Corin* KO mice, but not in *Corin* hcKO mice lacking cardiac corin, on 0.3% and 4% NaCl diets. Similar results of reduced Na^+^ and Cl^−^ excretion in the eccrine sweat glands were found in *Corin* KO, but not in *Corin* hcKO, mice [42]. It is known that hypertension can alter intestine structure and function [62]. In principle, the decreased fecal Na^+^ and Cl^−^ excretion in *Corin* KO mice could be caused by high blood pressure. As reported previously [47], both *Corin* KO and hcKO mice had increased blood pressure compared to that in WT mice on either a 0.3% or 4% NaCl diet. It is unlikely, therefore, that blood pressure level was a major determinant of fecal Na^+^ and Cl^−^ excretion in the mouse models tested in this study. Our findings support the idea that sodium excretion in the kidney, eccrine sweat glands, and intestine is promoted mainly by a local corin-mediated auto/paracrine mechanism, but not the cardiac corin-mediated endocrine mechanism.

Functional coupling between the intestine and the kidney is a compensatory mechanism to maintain homeostasis under pathophysiological conditions [63,64]. In chronic kidney failure, for example, decreased renal function is associated with increased fecal excretion of metabolites and electrolytes [65,66,67]. Conversely, increased fecal sodium loss is compensated by decreased urinary sodium excretion in mice with colon-specific ENaC deletion, particularly when dietary salt intakes are low [68]. In this study, we observed increased fecal Na^+^ and Cl^−^ excretion in *Corin* kcKO mice lacking renal corin. Such an increase in fecal Na^+^ and Cl^−^ excretion likely reflects a similar intestine–kidney compensatory mechanism to maintain sodium homeostasis in response to renal corin deficiency. Additional studies will be important to analyze urine aldosterone levels in these mouse models and to elucidate the molecular signaling mechanism underlying the intestine–kidney crosstalk.

Aldosterone is a central hormone for sodium reabsorption in the kidney, eccrine sweat glands, and colon [7,69,70]. One of the mechanisms underlying aldosterone function is to increase ENaC expression and/or activity, thereby promoting sodium reabsorption [69,71,72]. The inhibition of renal ENaC activity by ANP promotes natriuresis in the kidney [8,73]. In mice, corin deficiency increases ENaC expression/activity in the kidney and eccrine sweat glands, resulting in decreased sodium excretion [23,42,48]. Such a defect was ameliorated when the mice were administered amiloride, an ENaC inhibitor [23,42]. As shown in this study, fecal Na^+^ and Cl^−^ levels decreased when WT and *Corin* kcKO, but not *Corin* KO, mice were administered aldosterone, resulting in similar fecal Na^+^ and Cl^−^ levels among those three groups of mice on 0.3% and 4% NaCl diets. These results suggest that intestinal corin, present in WT and *Corin* kcKO mice, may have a similar function in antagonizing aldosterone action, which could be inhibited by exogeneous aldosterone. In contrast, such an aldosterone-antagonizing function of intestinal corin is lost in *Corin* KO mice. As a result, exogeneous aldosterone did not reduce fecal Na^+^ and Cl^−^ excretion in *Corin* KO mice. More investigations in additional animal models and perhaps in humans should help verify the observed intestinal corin function in regulating sodium homeostasis.

## 5. Conclusions

While most sodium excretion occurs in the kidney, colonic sodium excretion represents an important regulatory mechanism in sodium homeostasis, particularly when kidney function is compromised. Corin is responsible for ANP activation. Here, we show that corin, ANP, and Npra are expressed in the intestinal tract in mice. By analyzing mouse models with the *Corin* gene disrupted globally or selectively in the heart or kidney, we show that intestinal corin has a critical function in promoting fecal sodium and chloride excretion. This corin function is likely mediated via a paracrine mechanism to antagonize aldosterone action in the intestine. These findings help elucidate the hormonal mechanism controlling sodium excretion in the intestinal tract.

## Figures and Tables

**Figure 1 biology-12-00945-f001:**
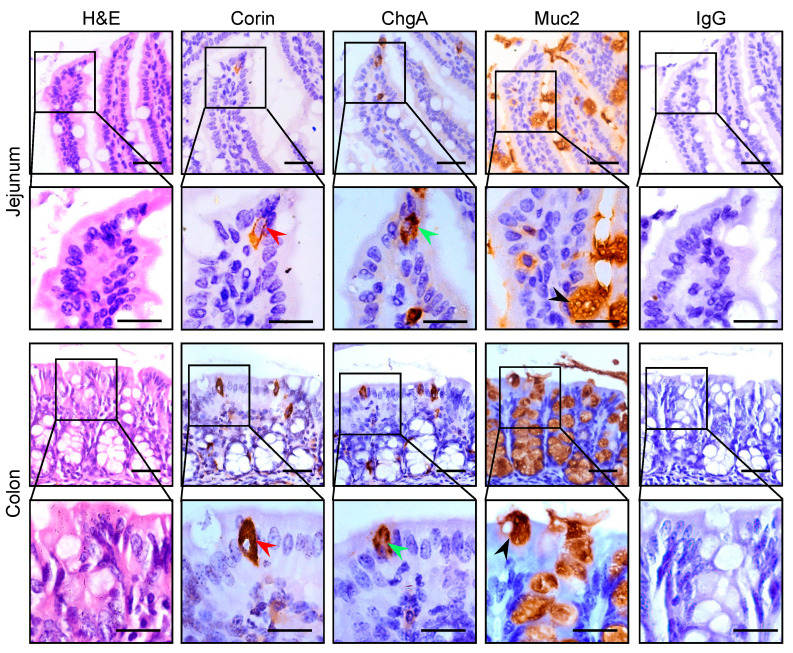
Corin protein expression in the jejunum and colon. Jejunal and colonic sections from WT mice (male, 10–12 weeks old on normal-salt diet) were stained with H&E (**left column**). Immunohistochemistry was performed using antibodies against corin (**2nd column**), ChgA (for enteroendocrine cells) (**3rd column**), and Muc2 (for goblet cells) (**4th column**). In the negative control, the primary antibody was replaced by normal IgG (**right column**). Red (**2nd column**), green (**3rd column**), and black (**4th column**) arrowheads indicate corin-, ChgA-, and Muc2-positive cells, respectively. Scale bars: 20 μm. Representative data were from at least three experiments in each group.

**Figure 2 biology-12-00945-f002:**
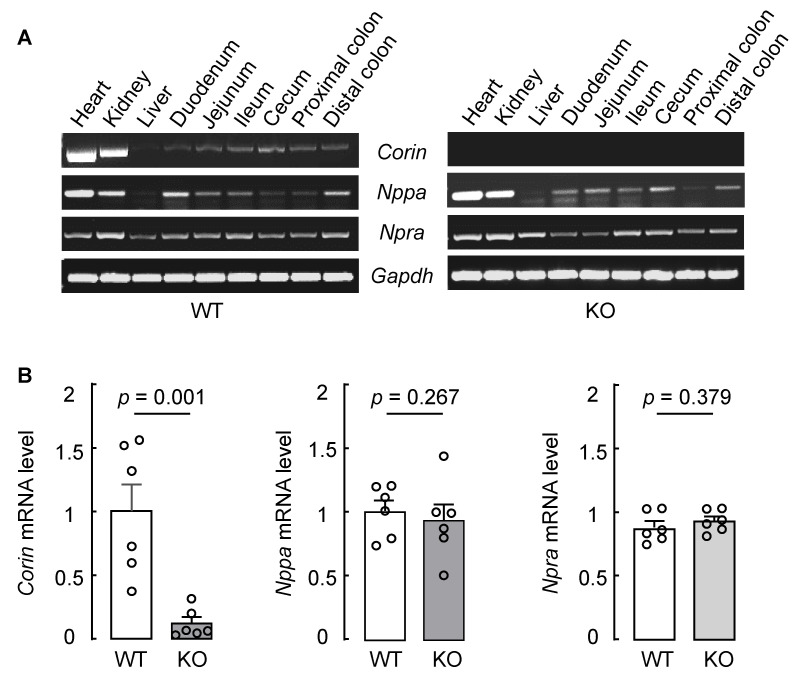
*Corin*, *Nppa*, and *Npra* mRNA expression in mouse intestinal segments. (**A**) Heart, kidney, liver, and intestine tissues from WT and *Corin* KO mice (male, 10–12 weeks old on normal-salt diet) were used to extract mRNAs. RT-PCR was performed to analyze *Corin*, *Nppa*, and *Npra* mRNA expression in intestinal segments, including the duodenum, jejunum, ileum, cecum, and colon. Heart and kidney samples were used as positive controls. Liver samples were used as a negative control. *Gapdh* mRNA level was analyzed as another control. Data are representative (*n* = 3). (**B**) Quantitative RT-PCR was conducted to assess *Corin*, *Nppa*, and *Npra* mRNA levels in mouse colon samples (*n* = 6 per group). Each dot represents a value obtained from an individual mouse. Data were examined by unpaired Student’s *t* test.

**Figure 3 biology-12-00945-f003:**
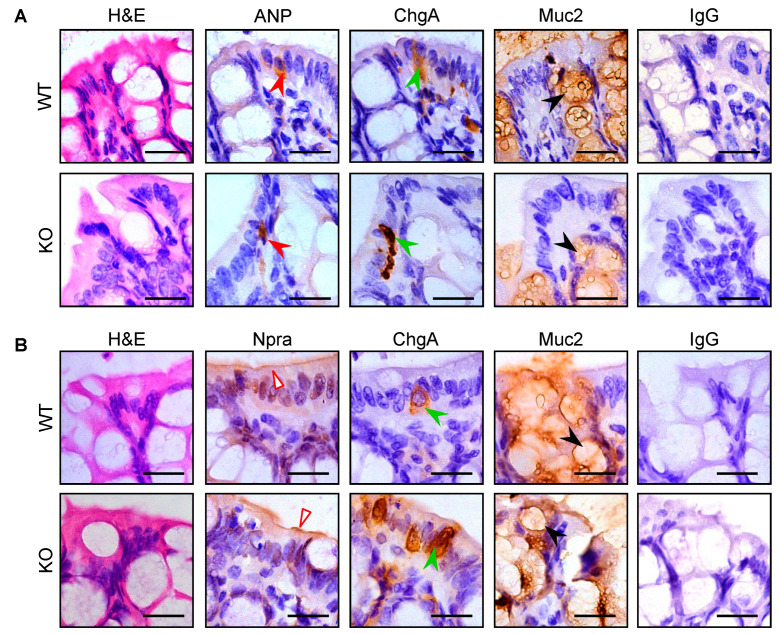
ANP and Npra expression in colon sections from WT and *Corin* KO mice. Colon samples were from WT and *Corin* KO mice (male, 10-12 weeks old on normal-salt diet). Immunohistochemical staining was conducted to identify pro-ANP/ANP (ANP) (**A**) (filled red arrowheads in rows 1–2) and Npra (**B**) (open red arrowheads in rows 3–4) protein expression in colonic sections form WT (rows 1 and 3) and *Corin* KO (rows 2 and 4) mice. ChgA (green arrowheads in 3rd column) and Muc2 (black arrowheads in 4th column) staining were included as controls. In the negative control, the primary antibody was replaced by normal IgG. Scale bars: 20 μm. Data are representative of at least three experiments in each group.

**Figure 4 biology-12-00945-f004:**
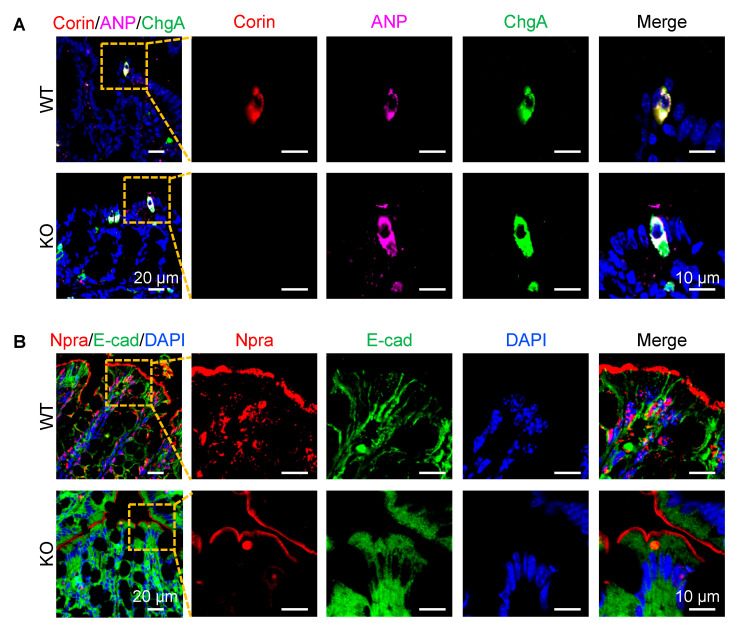
Corin, ANP, and Npra protein expression in the mouse colon. Co-immunofluorescent staining was performed in colonic sections from WT and *Corin* KO mice (male, 10–12 weeks old on normal-salt diet). (**A**) Co-staining of corin (red), ANP (purple), and ChgA (green) in enteroendocrine cells of colonic sections from WT (top) and *Corin* KO (bottom) mice. (**B**) Co-staining of Npra (red) and E-cad (green) on the luminal surface in colonic sections from WT (top) and *Corin* KO (bottom) mice. DAPI was used to stain cell nuclei (blue). Scale bars: 20 μm in column 1 and 10 μm in columns 2–5. Representative data were from at least three experiments in each group.

**Figure 5 biology-12-00945-f005:**
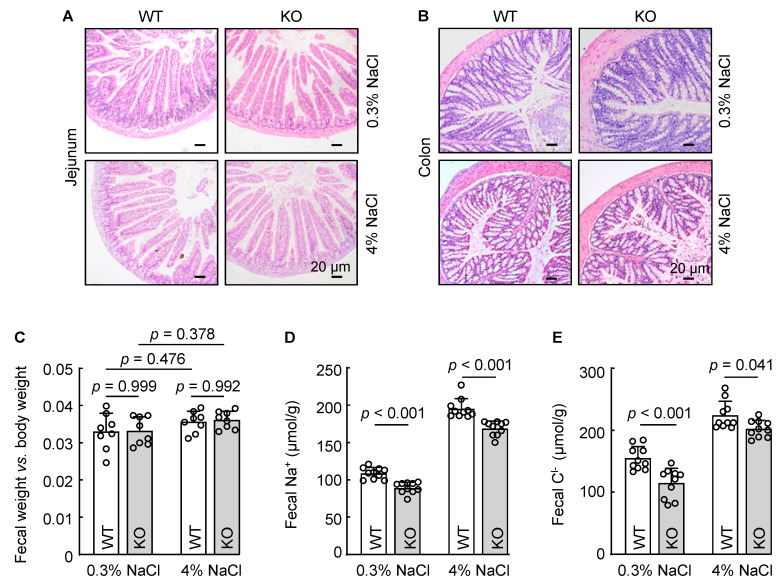
Analysis of intestinal histology, fecal weights, and fecal Na^+^ and Cl^−^ excretion in WT and *Corin* KO mice. (**A**,**B**) H&E staining of jejunal (**A**) and colonic (**B**) sections from WT and *Corin* KO mice (male, 10–12 weeks old) on 0.3% and 4% NaCl diets. Scale bars: 20 μm. Data are representative of three experiments. (**C**) Feces were collected for 24 h and dried. Data of dried fecal weights, normalized to body weights, were compared using one-way ANOVA. (**D**,**E**) Fecal Na^+^ (**D**) and Cl^−^ (**E**) levels in WT and *Corin* KO mice on 0.3% and 4% NaCl diets were measured. Each dot represents a value obtained from an individual mouse. *n* = 10 per group. Data are mean ± SEM. *p* values were examined with two-way ANOVA followed with Bonferroni multiple comparison test.

**Figure 6 biology-12-00945-f006:**
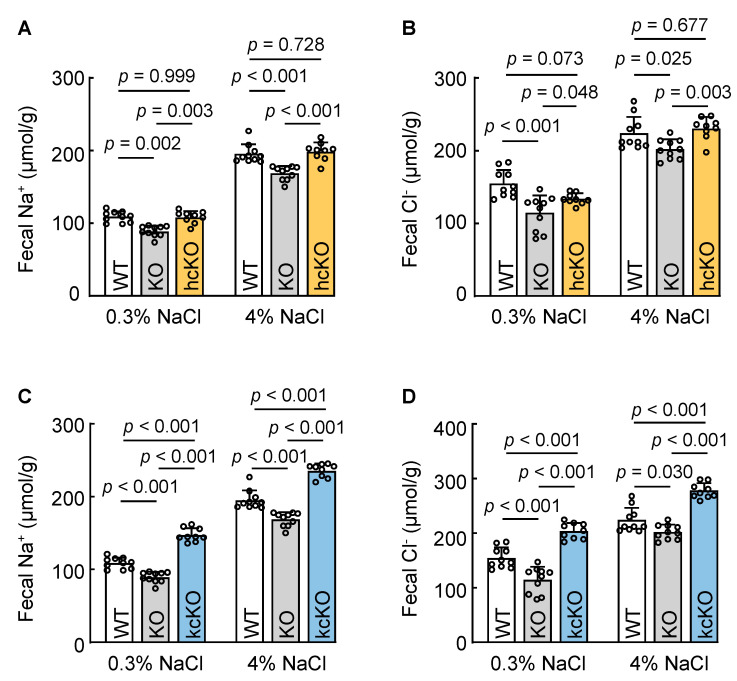
Fecal Na^+^ and Cl^−^ levels in WT, *Corin* KO, hcKO, and kcKO mice on 0.3% and 4% NaCl diets. Feces from WT, *Corin* KO, hcKO, and kcKO mice (male, 10–12 weeks old) on 0.3% and 4% NaCl diets were collected. Na^+^ and Cl^−^ concentrations were measured. (**A**,**B**) Fecal Na^+^ (**A**) and Cl^−^ (**B**) levels in WT, *Corin* KO, and *Corin* hcKO mice on 0.3% and 4% NaCl diets. *n* = 9–10 per group. (**C**,**D**) Fecal Na^+^ (**C**) and Cl^−^ (**D**) levels in WT, *Corin* KO, and *Corin* kcKO mice on 0.3% and 4% NaCl diets. Each dot represents a value obtained from an individual mouse. *n* = 9–10 per group. Data are mean ± SEM. *p* values were examined with two-way ANOVA and Turkey multiple comparison test.

**Figure 7 biology-12-00945-f007:**
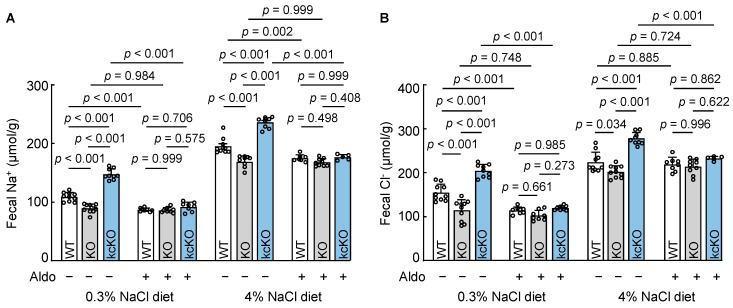
Effects of aldosterone on fecal Na^+^ and Cl^−^ excretion in WT, *Corin* KO, and kcKO mice. The mice (male, 10-12 weeks old) on 0.3% or 4% NaCl diets were administered aldosterone (Aldo). Fecal samples were collected for 24 h and Na^+^ and Cl^−^ levels were analyzed. Fecal Na^+^ (**A**) and Cl^−^ (**B**) levels in WT, *Corin* KO, and kcKO mice on 0.3% or 4% NaCl diets without (–) or with (+) aldosterone treatment. Each dot represents a value obtained from an individual mouse. *n* = 5–10 per group. Data of mean ± SEM are presented. *p* values were determined by two-way ANOVA with Turkey multiple comparison test.

## Data Availability

All data that support the findings of this study are presented in the manuscript and the Appendix A of this article. Data sharing is not applicable to this article.

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
