# Peer review of "Corin Deficiency Diminishes Intestinal Sodium Excretion in Mice"

_biology, 2023, doi:10.3390/biology12070945_

Round 1

Reviewer 1 Report

The manuscript investigated the expression and function of corin, ANP, and Npra in the mouse intestine, with a focus on their role in regulating sodium excretion. The authors found that corin is expressed in the almost all intestinal segments, and co-localized with ANP in enteroendocrine cells, while Npra was expressed on the luminal epithelial surface. Corin KO exhibited a decrease in fecal Na+ and Cl- excretion compared to WT mice. Cardiac specific Corin KO mice did not show changes in fecal Na+ and Cl- excretion. Renal specific Corin KO mice had increased fecal Na+ and Cl- excretion. The trends stood when the mice were switched from normal diet to high salt diet. When WT, Corin KO, and the Renal specific Corin KO were treated with aldosterone, the differences in fecal Na+ and Cl- levels disappeared. The authors utilized three different animal models, designed experiments carefully, provided clear and enough controls, and drew conclusions with solid background information from relevant literatures. The results are novel and meaningful in understating hormonal mechanism in controlling sodium excretion the intestinal tract. Here are some minor concerns:

1. Please provide the source of the animals, especially the conditional KO models in a clear way. It will be super helpful if the authors can briefly describe how the models were constructed.

2. Why not test the Na+ and Cl- excretion directly in Corin intestine KO mice?

3. Have you compared the ANP and ENaC expression level/activity in intestine of Corin KO mice? As the authors mentioned, AHP and ENaC are the keys for aldosterone function and sodium excretion; and aldosterone antagonizing function of intestinal corin is lost in Corin KO mice. A simple experiment like western blot can provide some hints for the hypotheses. 

N/A

Author Response

1. Please provide the source of the animals, especially the conditional KO models in a clear way. It will be super helpful if the authors can briefly describe how the models were constructed.

Response: We thank the reviewer for the suggestion. The Methods section has been revised to include more information regarding how these Corin-deficient mice were made (page 5, section 2.1. Mouse Models).

2. Why not test the Na+ and Cl- excretion directly in Corin intestine KO mice?

Response: This is a good question. Currently, Corin intestine KO mice are not available. As shown in our study, corin is expressed in a specific set of enteroendocrine cells. The transcriptional control of Corin expression in the enteroendocrine cells is unknown. We do not know what strains of Cre mice may be used to generate intestine specific Corin KO mice.

3. Have you compared the ANP and ENaC expression level/activity in intestine of Corin KO mice? As the authors mentioned, AHP and ENaC are the keys for aldosterone function and sodium excretion; and aldosterone antagonizing function of intestinal corin is lost in Corin KO mice. A simple experiment like western blot can provide some hints for the hypotheses.

Response: We thank the reviewer for the question. We have addressed the issue regarding the ANP and ENaC expression levels in intestine tissues in Corin KO mice. By quantitative RT-PCR, we showed that Nppa (encoding the ANP precursor) (Fig. 2B) and Scnn1a (encoding aENaC) (Fig. S3) expression levels were similar in intestine samples between WT and Corin KO mice.

Reviewer 2 Report

Dear authors,

I found your paper interesting and well designed with appropriate controls.

Nevertheless, I have some questions/comments regarding your paper in the aim to improve your manuscript. Please find it below:

 Line 55-56, the authors mentioned: “In longitudinal human genetic studies, CORIN variants are linked to dietary salt-sensitivity, blood pressure variations, and risk of hypertension”. But the blood pressure was not check in this study. After 3 weeks under 4% high salt diet it might be possible that these mice became hypertensive. Even if it is not their major aim, why the authors did not check the blood pressure? Hypertension can affect their results regarding sodium uptake in the intestine. Indeed, recent link between the gastrointestinal track and the blood pressure (PMID: 31236708). Besides, and it is a really nice and positive point, they check for Scnn1a gene. Thus, I do not understand why they did not checked for the blood pressure. They probably have to explain this point in the manuscript.

 Lines 87-88: “Most experiments were conducted in male mice (10-12 weeks old) that were randomized into test groups”. The mice used in this study are older than mice usually used in salty diet experiment. Indeed, as shown in these some recents examples, mice are fed with salty diet from 6 to 8 weeks-old (PMID: 32265505, PMID: 33845849, PMID: 32032975). Of course all the data obtained in mutated mice were compared with the data obtained in control mice. Nevertheless, are the authors not afraid that the age can have an effect to their results? Do they have references as examples to explain why they took quite old mice?

Line 168: “In negative controls, in which normal IgG, instead of specific antibodies”. It is not clear what you used as negative control. Did you used only the secondary antibody as it is usually done or something different? Please, detail more.

Figures 1 and 4: In the above pictures the scale is not visible. Figure 3: it is the inverse.

Regarding the results showed in figures 1 to 4 I think that the observations have been done before the salty diet but is not clear. Please write clearly somewhere that you first showed results from normal diet. Then, in the legends the ages of the rats are also not clear. Indeed, it is not ranges but precise age: 10 weeks. But in the methods part you mentioned: 10-12 weeks-old. Why this gap? Are the rats let alone in standard cage for 1 or 2 weeks before the salty diet?

Line 259-260: “Moreover, we found similar dry fecal weights, normalized to body weights, between WT and Corin KO mice on 0.3% NaCl diet”. I know that it is not the major aim of the article but what the results found with feces are similar in urine? It might be interesting to know.

For the results part 3.8, why the authors did not quantify aldosterone in the urine? Indeed, in the discussion they noted: “Functional coupling between the intestine and the kidney is a compensatory mechanism to maintain homeostasis under pathophysiological conditions”. Thus, it might be interesting to further explored the relationship between the kidney and the gastrointestinal tract.

Author Response

Line 55-56, the authors mentioned: “In longitudinal human genetic studies, CORIN variants are linked to dietary salt-sensitivity, blood pressure variations, and risk of hypertension”. But the blood pressure was not check in this study. After 3 weeks under 4% high salt diet it might be possible that these mice became hypertensive. Even if it is not their major aim, why the authors did not check the blood pressure? Hypertension can affect their results regarding sodium uptake in the intestine. Indeed, recent link between the gastrointestinal track and the blood pressure (PMID: 31236708). Besides, and it is a really nice and positive point, they check for Scnn1a gene. Thus, I do not understand why they did not checked for the blood pressure. They probably have to explain this point in the manuscript.

Response: This is a good point. Indeed, hypertension can alter intestine structure and function. In principle, the decreased fecal Na+ and Cl excretion in Corin KO mice could be caused by high blood pressure. In this study, decreased fecal Na+ and Cl levels were found in Corin KO mice, but not in Corin hcKO mice lacking cardiac corin, on 0.3% and 4% NaCl diets. As we reported previously (ref. 47), both Corin KO and hcKO mice had increased blood pressure compared to that in WT mice on either 0.3% or 4% NaCl diet. It is unlikely, therefore, that blood pressure level was a major determinant of fecal Na+ and Cl excretion in the mouse models tested in this study. We have revised the Discussion (page 14, second paragraph, lines 10-15) to reflect these points. The reference on hypertension and intestine function has been added (ref. 62).

Lines 87-88: “Most experiments were conducted in male mice (10-12 weeks old) that were randomized into test groups”. The mice used in this study are older than mice usually used in salty diet experiment. Indeed, as shown in these some recents examples, mice are fed with salty diet from 6 to 8 weeks-old (PMID: 32265505, PMID: 33845849, PMID: 32032975). Of course all the data obtained in mutated mice were compared with the data obtained in control mice. Nevertheless, are the authors not afraid that the age can have an effect to their results? Do they have references as examples to explain why they took quite old mice?

Response: We thank the reviewer for pointing this out. We did use 6-8-week-old mice at the starting point. After three weeks of salt diet treatment, the mice became 9-11 weeks old. Additional days were required to collect fecal samples, extending the age to 10-12 weeks. Indeed, the original description regarding mouse ages was not accurate. It has been revised accordingly (page 6, line 6). Changes have also been made in figure legends. As the reviewer indicated, we did use age-matched WT mice as controls in our experiments. 

Line 168: “In negative controls, in which normal IgG, instead of specific antibodies”. It is not clear what you used as negative control. Did you used only the secondary antibody as it is usually done or something different? Please, detail more.

Response: Thank you for asking to clarify. In this negative control, the primary antibody was replaced by normal IgG. All sections were incubated with a secondary antibody. We have revised the text in the Methods (page 6, last 3 lines) and the Results (page 9, first paragraph, lines 6-7).

Figures 1 and 4: In the above pictures the scale is not visible. Figure 3: it is the inverse.

Response: Indeed, the scale bar labeling was not visible, due to the dark background. We have removed the labeling in the figures and provided the information in the figure legend.

Regarding the results showed in figures 1 to 4 I think that the observations have been done before the salty diet but is not clear. Please write clearly somewhere that you first showed results from normal diet. Then, in the legends the ages of the rats are also not clear. Indeed, it is not ranges but precise age: 10 weeks. But in the methods part you mentioned: 10-12 weeks-old. Why this gap? Are the rats let alone in standard cage for 1 or 2 weeks before the salty diet?

Response: As indicated above, mice were 6-8 weeks old at the starting point. Some tissue samples were collected from the mice after 3 weeks of normal-salt diet and additional days for fecal analysis. We have revised the figure legends to indicate the gender and age of mice and specific diet in each experiment.

Line 259-260: “Moreover, we found similar dry fecal weights, normalized to body weights, between WT and Corin KO mice on 0.3% NaCl diet”. I know that it is not the major aim of the article but what the results found with feces are similar in urine? It might be interesting to know.

Response: Thank you for asking. As we reported previously, urine volumes in WT and Corin KO mice on 0.3% NaCl diet were similar (Fig. 6C in ref. 47). As the reviewer indicated, this is not a major point of this study, so we did not discuss it in the manuscript.

For the results part 3.8, why the authors did not quantify aldosterone in the urine? Indeed, in the discussion they noted: “Functional coupling between the intestine and the kidney is a compensatory mechanism to maintain homeostasis under pathophysiological conditions”. Thus, it might be interesting to further explored the relationship between the kidney and the gastrointestinal tract.

Response: This is a good point. In previous studies, reduced serum aldosterone levels were found in both Corin KO and kcKO mice, compared to that in WT mice (Fig. 2D in ref. 47). Similar findings of reduced serum aldosterone levels were reported in another strain of Corin KO mice (Fig. 4b in ref. 23). In this study, fecal Na+ and Cl- excretion decreased in Corin KO mice but increased in Corin kcKO mice. The results support the hypothesis that the role of intestine corin in antagonizing aldosterone action in the regulation of fecal sodium excretion is local but not systemic. To reflect the reviewer’s point, we have revised the Discussion to indicate that “Additional studies will be important to analyze urine aldosterone levels in these mouse models. . .” (page 15; first paragraph, last three lines).

Reviewer 3 Report

he current study handles the the actual role of  intestinal corin in maintaining sodium homeostasis with emphasis of its antagonizing role to aldosterone.

the author should address the following comments: 

1-     Line 25 were co-localized in stead of was co- localized

2-     Line 77 , the authors should not point out their results in the aim of the study.

3-     The number of mice per group should be mentioned in materials and methods section as well as the age of mice.

4-     The composition diet of experimental animals should be pointed out.

5-     Results should not contain citations. Just only mention the obtained results.

minor English editing is required

Author Response

1. Line 25 were co-localized in stead of was co-localized.

Response: We thank the reviewer for critical reading. If we understand well, the subject of the sentence is ‘expression’, which is singular but not plural. Thus, the word ‘was’ was used. If this is incorrect, we will be happy to revise the sentence.

2. Line 77, the authors should not point out their results in the aim of the study.

Response: This is a good point. We have revised the text accordingly (page 5, second paragraph, last two lines).

3. The number of mice per group should be mentioned in materials and methods section as well as the age of mice.

Response: Thank you for the suggestion. We have revised the Materials and Methods to provide the information on the number of mice (page 6, line 6). Figure legends have also been revised to provide information on the gender and age of the mice and specific diet.

4. The composition diet of experimental animals should be pointed out.

Response: As suggested, we have added this information in the Materials and Methods section (page 6, lines 2-5).

5. Results should not contain citations. Just only mention the obtained results.

Response: As suggested, the citations in the Results section have been removed.
